# Effects of a Self-Prepared Carbohydrate-Reduced High-Protein Diet on Cardiovascular Disease Risk Markers in Patients with Type 2 Diabetes

**DOI:** 10.3390/nu13051694

**Published:** 2021-05-17

**Authors:** Ahmad H. Alzahrani, Mads J. Skytte, Amirsalar Samkani, Mads N. Thomsen, Arne Astrup, Christian Ritz, Jan Frystyk, Jens J. Holst, Sten Madsbad, Steen B. Haugaard, Thure Krarup, Thomas M. Larsen, Faidon Magkos

**Affiliations:** 1Department of Nutrition, Exercise and Sports, University of Copenhagen, 1958 Frederiksberg, Denmark; ahmad@nexs.ku.dk (A.H.A.); ara@novo.dk (A.A.); ritz@nexs.ku.dk (C.R.); krarupthure@gmail.com (T.K.); thomaslas@gmail.com (T.M.L.); 2Department of Physiology, Faculty of Medicine, The University of Jeddah, Jeddah 23218, Saudi Arabia; 3Department of Endocrinology, Copenhagen University Hospital Bispebjerg, 2400 Copenhagen, Denmark; madsskytte@hotmail.com (M.J.S.); amsam03@gmail.com (A.S.); madsnorvinthomsen@gmail.com (M.N.T.); d299057@dadlnet.dk (S.B.H.); 4Department of Clinical Medicine, Health, Aarhus University, 8000 Aarhus, Denmark; jan.frystyk@rsyd.dk; 5The Research Unit for Endocrinology, Department of Endocrinology, Odense University Hospital, 5000 Odense, Denmark; 6Novo Nordisk Foundation Center for Basic Metabolic Research, Department of Biomedical Sciences, University of Copenhagen, 2200 Copenhagen, Denmark; jjholst@sund.ku.dk (J.J.H.); Sten.Madsbad@regionh.dk (S.M.); 7Department of Internal Medicine, Copenhagen University Hospital Amager Hvidovre, 2650 Hvidovre, Denmark

**Keywords:** low-carbohydrate, high-protein, lipids, cardiovascular

## Abstract

We previously observed beneficial effects of a carbohydrate-reduced, high-protein (CRHP) diet on cardiovascular risk markers in patients with type 2 diabetes mellitus (T2DM) in a crossover 2 × 6-week trial, when all food was provided to subjects as ready-to-eat meals. Here, we report the results from a 6-month open label extension: 28 patients with T2DM were instructed to self-prepare the CRHP diet with dietetic guidance. At weeks 0, 6, 12, and 36, fasting and postprandial (4-h meal test) blood samples were collected for measurements of total, high-density lipoprotein (HDL) and low-density lipoprotein (LDL) cholesterol, triacylglycerol (TG), apolipoproteins A1 and B, non-esterified fatty acids (NEFA), C-reactive protein (CRP), tumor necrosis factor-α (TNF-α), and interleukin-6. Diurnal blood pressure and heart rate were also assessed. At the end of the study (week 36), concentrations of fasting total and LDL-cholesterol, fasting and postprandial NEFA and TG, and fasting apolipoprotein-B, CRP and TNF-α concentrations were significantly lower compared with week 0 (*p* < 0.05). A significant decrease in diurnal heart rate was also observed. From week 12 to 36, an increase in HDL-cholesterol and apolipoprotein-A1 concentrations and a further reduction in fasting and postprandial NEFA (*p* < 0.05) were found. These changes were independent of minor fluctuations in body weight. We conclude that the substitution of dietary carbohydrate for protein and fat has beneficial effects on several cardiovascular risk markers in patients with T2DM, which are maintained or augmented over the next 6 months when patients select and prepare the CRHP diet on their own in a dietitian-supported setting.

## 1. Introduction

Type 2 diabetes mellitus (T2DM) is a major public health problem that is approaching epidemic proportions. The global prevalence among adults over 18 years of age has risen from 4.3–5.0% in 1980 to 7.9–9.0% in 2014 [1]. About 1 in 11 adults has diabetes (463 million) and ~10% of global healthcare expenditure is spent on diabetes management [2]. Patients with T2DM have increased total mortality, with cardiovascular disease (CVD) being one of the leading causes of death [3]. This is due to the multiple abnormalities in lipid metabolism [4], including increased circulating cholesterol [5], triacylglycerol (TG) [6], and non-esterified fatty acid (NEFA) concentrations in the fasting and postprandial states [7], increased resting and diurnal blood pressure [8,9], and increased markers of inflammation [10].

T2DM it tightly linked to the increasing body mass index (BMI) and obesity [11]. Accordingly, weight loss is the cornerstone of T2DM lifestyle management [12], but scientific evidence for the optimal dietary treatment of T2DM is lacking. The most recent American Diabetes Association recommendations suggest low-carbohydrate and very-low-carbohydrate diets as a viable treatment strategy for some individuals with T2DM [13], even if these diets do not necessarily lead to greater weight loss compared with other hypocaloric regimens [14,15]. Carbohydrate-reduced, high-protein (CRHP), non-ketogenic diets induce beneficial effects on glucose control in individuals with T2DM even in the absence of weight loss, and are thus being advocated to ameliorate hyperglycemia [16,17,18,19]. We recently reported the results of a randomized crossover trial (2 × 6-weeks) in which we evaluated the effects of a weight-maintaining CRHP diet compared with an isoenergetic control diet in patients with T2DM [20,21]. In that study, the CRHP diet brought about a number of beneficial changes in the CVD risk factor profile [20,21], but since all food was prepared by the investigating team in a metabolic kitchen and provided to the subjects free of charge as ready-to-eat meals (ensuring near-complete control over the dietary intake and a high level of compliance), it remains unknown whether these benefits can be maintained in the longer term, under more real-life and less resource-demanding conditions.

Accordingly, in the present study, we followed our patient cohort for another 6 months. During this period, all the participants were given instructions to help them keep consuming a weight-maintaining diet similar to the CRHP diet they received during the highly-controlled 2 × 6-week intervention phase [20,21], but now they were required to purchase the food themselves and prepare the CRHP diet on their own, in a dietitian-supported setting. We hypothesized that the beneficial changes in CVD risk factors (i.e., lipid profile, inflammatory markers, blood pressure, and heart rate) that were observed after 6 weeks on the fully-provided CRHP diet [20,21] would be maintained or further improve during the 6 months of follow-up when all the subjects would consume a self-prepared CRHP diet.

## 2. Materials and Methods

### 2.1. Participants

Men and women with T2DM were recruited on the basis of the following inclusion criteria: T2DM diagnosis [21,22]; HbA1c 6.5–11% (48–97 mmol/mol); age > 18 years; hemoglobin > 1 g/L for women and >1.1 g/L for men; and an estimated glomerular filtration rate > 30 mL/min/1.73 m^2^ (actual minimum was ~56). Exclusion criteria included critical illness; systemic corticosteroid treatment; severe food allergy or intolerance; severe gut disease; alcohol dependence syndrome; injectable diabetes medication; repeated fasting plasma glucose > 13.3 mmol/L; urine albumin/creatinine ratio > 300 mg/g; lactation, pregnancy or planning of pregnancy during the study; and inability to comply with the study procedures [20,21].

We report here on a 6-month prospective follow-up of our previous open-label, randomized, crossover, 2 × 6-week trial [20,21]. All the patients signed an informed consent prior to participating in the study, which was approved by the regional Ethical Committee of Copenhagen and carried out in accordance with the Helsinki declaration. The study is registered at clinicaltrials.gov (NCT02764021) and was conducted between April 2016 and November 2017 (inclusive of follow-up).

### 2.2. Study Design

During the initial 12 weeks (2 × 6-week dietary periods), patients were provided with pre-packaged ready-to-eat food that met their daily energy needs for weight maintenance. They were restricted to consume only the meals and beverages provided. The ad libitum intake of non-caloric beverages was allowed, but alcohol consumption was prohibited. The amount of energy of the fully-provided diet was adjusted periodically to ensure weight stability. Consistent with European dietary guidelines [23], the control diet comprised 50% of energy as carbohydrate, 17% as protein, and 33% as fat. By contrast, the CRHP diet comprised 30% of energy as carbohydrate, 30% as protein, and 40% as fat [21]. Meals were prepared in the metabolic kitchen of the Department of Nutrition, Exercise and Sports at the University of Copenhagen, as a 7-day rotational menu, and distributed to participants twice weekly. The self-prepared diet phase started immediately after the end of the 12-week fully-provided diet phase and lasted 6 months, i.e., from week 12 to week 36. Subjects were instructed to consume a self-selected and self-prepared CRHP diet based on their experience during the controlled phase of the study. They were encouraged to limit alcohol consumption and maintain the same level of habitual physical activity throughout the study. To ensure that participants would be able to self-select and self-prepare a CRHP diet during the follow-up, they met with the study dietitian every 2 weeks to review the progress, facilitate compliance to the CRHP diet, and help maintain body weight. This was deemed necessary as the full provision of the diet as ready-to-eat meals during the first, controlled phase of the study would not make them very familiar with the raw food items and ingredients they would need to purchase in the second, follow-up phase. Participants were also provided with recipe books and were instructed to record food intake for 3 days (2 weekdays and 1 weekend day) during weeks 19, 25, and 32 using the online dietary reporting tool MADLOG [24]. The diet analysis was conducted immediately, and patients were advised accordingly by the dietitian towards meeting the macronutrient targets.

### 2.3. Outcome Assessment

All physical examination procedures and metabolic function tests were performed at baseline (week 0), after each 6-week period on the CRHP and control diets (at 6 and 12 weeks), and at the end of the study (36 weeks).

#### 2.3.1. Fasting CVD Risk Factor Profile

Blood samples were collected after a 12-h overnight fast to separate plasma or serum. Serum was used for the analysis of NEFA, total cholesterol, high density lipoprotein (HDL) cholesterol, TG, apolipoprotein A1 (apoA1), and apolipoprotein B (apoB). The Wako NEFA-HR(2) kit (ASC-ACOD method, Wako Chemicals GmbH, Neuss, Germany) was used to determine NEFA, and enzymatic colorimetric assays (CHOL2, HDLC4, and TRIGL, Cobas 6000, Roche Diagnostics GmbH, Mannheim, Germany) were used to determine total cholesterol, HDL-cholesterol, and TG, respectively. The low-density lipoprotein (LDL) cholesterol was calculated using the Friedewald equation [25] and the very low-density lipoprotein (VLDL) cholesterol was calculated by subtraction (total cholesterol–HDL–LDL). ApoA1 and apoB were determined using immunoturbidimetric assays (APOAT and APOBT, respectively; Cobas 6000, Roche Diagnostics GmbH, Mannheim, Germany). The C-reactive protein (CRP) in serum was measured by the IMMULITE 2000 platform (High sensitivity CRP, Siemens Healthcare Diagnostics Products Ltd., Llanberis, UK). Plasma tumor necrosis factor-alpha (TNF-α) and interleukin-6 (IL-6) concentrations were measured by multi-spot immunoassays (V-PLEX, Meso Scale Discovery, Rockville, USA). For each analyte, all samples from each subject were analyzed in the same assay run.

#### 2.3.2. Postprandial Lipid Responses

After an overnight fast, a solid mixed meal with an energy content equal to ~25% of each subject’s total energy requirements (~600 kcal, with 30% of energy from carbohydrate, 30% from protein, and 40% from fat; 49 mg of cholesterol and 16 g fiber) was consumed within 25 min. The meal consisted of yoghurt (0.2% fat), blueberries, almonds, ham, butter, crispbread, cheese 30+ (the plus sign indicates the fat content in dry weight), and water. Blood samples were collected in the fasting state and for 4 h after meal ingestion to measure postprandial NEFA and TG concentrations. Meal-induced responses were calculated as total areas under the curve (AUC) using the trapezoidal rule [26].

#### 2.3.3. Diurnal Blood Pressure and Heart Rate

An ambulatory monitoring system (ABPM, Model 90217, Spacelabs Healthcare, Snoqualmie, WA, USA) was used to measure diurnal systolic and diastolic blood pressures (SBP and DBP, respectively), mean arterial pressure (MAP), pulse pressure, and heart rate. Daily profiles were summarized as total AUCs.

### 2.4. Statistical Analysis

The outcomes (primary, secondary, and exploratory) of this analysis were the same as those of the main study [21]. Given this was an open-label extension, no separate power calculation was made. Power calculations for sample size determination were based on the primary outcome (HbA1c) of the initial 2 × 6-week, fully-provided diet intervention phase, and have been described in detail previously [21]. The present study followed up on the participants for an additional 6-month period. Given that all 28 subjects completed the self-prepared diet period, the statistical power to detect changes in all outcomes against baseline was the same as that during the fully-provided diet phase.

A linear mixed effects model with time as a fixed effect was used to test for significant differences between values at the end of the study (week 36) against baseline (week 0) and against the end of the fully-provided CRHP diet. The order of diets during the initial 12 weeks of the study (half of the subjects received the control diet first, whereas the other half received the CRHP diet first) was included as a covariate in the model. Body weight was included as an additional covariate to evaluate whether the effects of the CRHP diet were driven by changes in body weight. Data were tested for normality according to Shapiro-Wilk and, if necessary, log-transformed to ensure normality. If no appropriate transformation could be found, ranked data were used for analysis. A *p*-value < 0.05 was considered significant. Results are presented as means ± SD for normally distributed data, back-transformed means with 95% confidence intervals for log-transformed data or medians with quartiles for ranked data. Statistical analyses were conducted with R version 3.6.1 and RStudio version 1.2.5019 (RStudio, Boston, MA, USA).

## 3. Results

### 3.1. Patient Characteristics and Self-Prepared Diet

A total of 28 patients (20 males and 8 females) with T2DM and a BMI between 21.6 and 39.4 kg/m^2^ completed the initial 2 × 6-week fully-provided diet intervention, and all were followed-up for 6 months (Table 1).

During the self-selected CRHP diet period, participants were consuming an average of 2322 ± 814 kcal/day with 28 ± 4% of energy from carbohydrate, 27 ± 4% from protein, 2 ± 1% from fiber, 1 ± 2% from alcohol, and 41 ± 5% from fat. Out of all fat calories, saturated, monounsaturated, and polyunsaturated fatty acids accounted for 41 ± 6%, 41 ± 5%, and 18 ± 3%, respectively.

### 3.2. Changes in Body Weight and Medications

All medications were kept constant during the fully-provided diet phase. During the subsequent 6 months, no changes in antidiabetic and lipid-lowering treatment regimens were recorded. Antihypertensive medications also did not change for the majority of patients except one, for whom an increase (Ancozan from 50 to 100 mg daily) occurred during follow-up.

Body weight at the end of the study (week 36: 86.3 ± 19.3 kg) was significantly lower (*p* < 0.001) compared with baseline (week 0: 89.1 ± 19.4 kg), but was not different compared with the end of the fully-provided diet period (86.9 ± 18.9 kg). Adjusting the analyses for body weight did not affect the statistical significance of the results for CVD risk markers.

### 3.3. Lipid and Inflammatory Marker Responses

At the end of the study (week 36), total cholesterol concentration, LDL and VLDL cholesterol concentrations, fasting and postprandial NEFA and TG concentrations, fasting apoB concentration and the apoB/apoA1 ratio, and fasting CRP and TNF-α concentrations were significantly lower compared with baseline (Table 2). Compared with the end of the fully-provided diet period, follow-up was accompanied by a significant increase in total cholesterol concentration, predominantly due to an increase in HDL-cholesterol coupled with an increase in apoA1 concentration (Table 2). In addition, further reductions in fasting and postprandial NEFA concentrations were observed (Table 2).

### 3.4. Diurnal Blood Pressure and Heart Rate Responses

There were no significant differences in the daily average values and 24-h AUCs for SBP, DBP, MAP, and pulse pressure, but heart rate was significantly lower at the end of the study compared with baseline. This occurred exclusively during the fully-provided diet phase with no further change during the 6-month follow-up (Table 3).

## 4. Discussion

In this study, we demonstrate that patients with T2DM can effectively self-select and self-prepare a CRHP diet, when they receive some dietitian support, in order to maintain and, in some instances, augment the beneficial effects on CVD risk factors obtained during an initial tightly-controlled dietary intervention, when all food was prepared by the investigators and provided to them free of charge. Patients were able to follow this diet on their own for the subsequent 6 months and, at the end of the follow-up, presented with considerable improvements in fasting blood lipid profile and postprandial lipemia, and small improvements in markers of inflammation and diurnal heart rate. These findings reinforce the utility of carbohydrate restriction for mitigating CVD risk factors in T2DM, even in the absence of major weight loss.

We observed that most of the beneficial effects on lipid metabolism that were observed during the initial 2 × 6-week period of the study [21], including significant reductions in total and LDL cholesterol concentrations, apoB concentration, and fasting and postprandial NEFA and TG concentrations, were maintained during the subsequent 6 months of follow-up. Importantly, fasting and postprandial NEFA concentrations decreased further, suggesting an even more efficient regulation of adipose tissue lipolysis, perhaps owing to improved adipose tissue insulin sensitivity both in the fasting state as well as following meal ingestion. Most of these results are in line with data from most [18,21,27] but not all [13] previous studies.

Interestingly, during the self-prepared dietary period, we also observed significant increases in HDL cholesterol and apoA1 concentrations compared with the end of the fully-provided diet period. These beneficial effects were not evident during the initial 2 × 6-week period of the study, nor were there any changes in the apoB/apoA1 ratio [21]. The effects of carbohydrate restriction on HDL metabolism are not entirely consistent, and may depend on concomitant changes in body weight and glycemic control. The conclusion of one meta-analysis of data from 13 studies (lasting from 3 months to 1 year) evaluating the metabolic effects of diet treatment in 1138 patients with T2DM suggested that low-carbohydrate high-protein diets do not improve HDL cholesterol but decrease TG concentration [28]. However, another meta-analysis of 23 trials (lasting from 3 months to 2 years) in 1357 participants with T2DM reported a significant decrease in TG concentration coupled with a near significant increase in HDL cholesterol after 6 and 12 months on a low-carbohydrate diet [14]. In addition, another meta-analysis of 19 shorter trials (lasting from 10 days to 6 weeks) in 306 patients with T2DM concluded that low-carbohydrate diets decrease TG and increase HDL cholesterol compared with low-fat diets [29]. Our findings are internally consistent as both HDL cholesterol and apoA1 increased, and may thus suggest that longer periods of time are needed for CRHP diet-induced improvements in HDL metabolism to manifest, perhaps due to the slow turnover rate of HDL particles. The increase in apoA1 concentration during the self-selected diet period resulted in the apoB/apoA1 ratio being significantly lower at the end of the study compared with baseline. This ratio is a measure related to atherogenic particle concentration (apoB is found in LDL and VLDL particles and apoA1 is found in HDL particles) and increased values are robust predictors of CVD [30,31].

Diurnal blood pressures did not change significantly after the CRHP during the initial 2 × 6-week fully-provided diet period [21], and we also did not observe any significant changes during the subsequent 6 months of follow-up. However, blood pressure was not particularly high in our patients (average resting systolic/diastolic blood pressure was 123/75 at baseline), which may have limited the efficacy of the CRHP diet. On the other hand, we found a significant decrease in diurnal heart rate (daily average and AUC), which is beneficial given that an increased heart rate (during resting, sleeping or throughout the day) has been associated with an increased total mortality and CVD risk among middle-aged and elderly men and women with no apparent heart disease [32].

Acute phase reactants correlate with inflammation in different ways and have been reported to have various important roles in the pathogenesis of cardiometabolic disorders including CVD [9,19]. For example, any intervention that reduces levels of inflammatory markers in the systemic circulation will likely be beneficial in controlling T2DM and cardiovascular complications [33]. In our study, fasting IL-6 concentrations did not change, but circulating CRP and TNF-α decreased at the end of follow-up. Reductions in TNF-α but not CRP were observed during the initial 2 × 6-week period of the study [20], suggesting that different markers of low-grade inflammation in these patients respond to the dietary treatment at a different time-frame, and certainly less robustly than the metabolic function.

Our study has several limitations, including lack of a control group, lack of objective assessment of physical activity, relatively small sample size and uneven male/female distribution, short duration of follow-up (6 months), and the fact that we had to statistically account for the crossover nature of the dietary intervention during the initial 12 weeks of the study (i.e., the order in which patients received the fully-provided CRHP diet). On the other hand, there were no dropouts during the follow-up period (from week 12 to 36) and patients complied to a large extent with the dietary instructions (as reflected by their dietary records and the maintenance of stable body weight). Our study was designed to mimic a real life setting where patients with T2DM self-select their own diet based on prior experience and dietitian guidance, and demonstrates the applicability and effectiveness of this approach to manage T2DM in practice, at least in the short-term. Future studies with larger sample sizes and longer follow-up periods are required to further assess the feasibility and sustainability of CRHP dietary regimens for patients with T2DM.

## 5. Conclusions

We conclude that the substitution of dietary carbohydrate for protein and fat has beneficial effects on several CVD markers in patients with T2DM, which are maintained or further improved over the next 6 months when patients select and prepare the CRHP diet on their own. These findings have important clinical implications, as they suggest that patients with T2DM can effectively follow low-carbohydrate high-protein diets, with some guidance from dietitians, and gain considerable benefits on a number of CVD risk factors in the absence of major weight loss.

## Figures and Tables

**Table 1 nutrients-13-01694-t001:** Baseline characteristics of patients with T2DM.

Variable	Value
Sex, male/female (*n* (%))	20/8 (71/29)
Age (years)	64 ± 7.7
Duration of T2DM (years)	7.0 ± 5.4
Body mass index (kg/m^2^)	30.1 ± 5.2
Fasting plasma glucose (mmol/L)	9.4 ± 1.4
HbA1c (mmol/mol)	59.6 ± 8.4
HbA1c (%)	7.6 ± 0.8
Medication use (*n* (%))	
No hypoglycemic agents	4 (14)
Using hypoglycemic medication	24 (86)
1 hypoglycemic agent	15 (54)
2 hypoglycemic agents	6 (21)
3 hypoglycemic agents	3 (11)
>3 hypoglycemic agents	0 (0)
Biguanide	22 (79)
DPP-4 inhibitors	9 (32)
SGLT2 inhibitors	5 (18)
Using lipid-lowering medication	20 (71)
Using antihypertensive medication	16 (57)

Values are shown as means ± SD or frequencies (percentages). T2DM: Type 2 diabetes mellitus; HbA1c: Glycated hemoglobin; DPP-4: Dipeptidyl peptidase-4; SGLT2: Sodium-glucose cotransporter 2.

**Table 2 nutrients-13-01694-t002:** Effects of a CRHP diet on lipids, apolipoproteins, and inflammatory markers in patients with T2DM.

	Baseline (Week 0)	End of Fully-Provided CRHP Diet (Week 6 or 12)	End of Self-Prepared CRHP Diet (Week 36)	P1	P2
Total cholesterol (mmol/L)	3.9 ± 0.9	3.4 ± 0.8	3.7 ± 0.9	0.003	0.045
HDL-cholesterol (mmol/L)	1.11 ± 0.24	1.07 ± 0.22	1.14 ± 0.25	0.196	0.003
LDL-cholesterol (mmol/L)	2.44 ± 0.85	2.04 ± 0.73	2.26 ± 0.85	0.009	0.062
VLDL-cholesterol (mmol/L)	0.34 (0.30, 0.39)	0.24 (0.21, 0.27)	0.24 (0.22, 0.28)	<0.001	0.247
NEFA, fasting (mmol/L)	0.77 (0.69, 0.85)	0.64 (0.58, 0.71)	0.46 (0.41, 0.51)	<0.001	<0.001
NEFA, 4-h AUC (pmol/L)	102 (92, 114)	81 (75, 88)	67 (60, 74)	<0.001	<0.001
Triacylglycerol, fasting (mmol/L)	1.71 (1.49, 1.97)	1.20 (1.07, 1.35)	1.22 (1.09, 1.38)	<0.001	0.247
Triacylglycerol, 4-h AUC (mmol/L)	441 (388, 501)	335 (298, 376)	344 (307, 386)	<0.001	0.188
ApoA1 (µmol/L)	49.1 ± 6.5	45.4 ± 6.3	48.5 ± 7.4	0.489	0.004
ApoB (µmol/L)	1.71 ± 0.43	1.45 ± 0.37	1.56 ± 0.41	<0.001	0.196
ApoB/ApoA1	0.036 ± 0.010	0.033 ± 0.009	0.033 ± 0.010	0.001	0.595
C-reactive protein (mg/L)	2.3 (1.5, 3.3)	1.6 (0.9, 2.5)	1.5 (0.9, 2.3)	0.016	0.771
Interleukin-6 (pg/mL)	0.8 (0.7, 1.1)	0.7 (0.6, 0.8)	0.8 (0.6, 0.9)	0.322	0.549
Tumor necrosis factor-α (pg/mL)	2.41 (2.08, 2.79)	2.22 (1.95, 2.53)	2.18 (1.91, 2.49)	<0.001	0.672

Data shown as mean ± SD or means and 95% confidence intervals. P1 = *p*-value for end of study (week 36) against baseline (week 0). P2 = *p*-value for end of study (week 36) against the end of fully-provided CRHP diet phase (week 6 or 12). CRHP: Carbohydrate-reduced high-protein; T2DM: Type 2 diabetes mellitus; HDL/LDL/VLDL: High-/low-/very-low density lipoprotein; NEFA: Non-esterified fatty acid; AUC: Area under the curve; Apo: Apolipoprotein.

**Table 3 nutrients-13-01694-t003:** Effects of a CRHP diet on blood pressure and heart rate in patients with T2DM.

	Baseline (Week 0)	End of Fully-Provided CRHP Diet (Week 6 or 12)	End of Self-Prepared CRHP Diet (Week 36)	P1	P2
Systolic BP, daily average (mmHg)	126 (122, 131)	120 (116, 125)	126 (122, 130)	0.811	0.144
Systolic BP, 24-h AUC (mmHg·h)	2646 (2553, 2743)	2511 (2417, 2608)	2636 (2556, 2719)	0.785	0.089
Diastolic BP, daily average (mmHg)	77 (73, 81)	74 (70, 79)	75 (72, 79)	0.262	0.418
Diastolic BP, 24-h AUC (mmHg·h)	1627 ± 199	1534 ± 181	1588 ± 159	0.082	0.344
Mean arterial pressure, daily average (mmHg)	94 ± 9	90 ± 9	93 ± 7	0.221	0.296
Mean arterial pressure, 24-h AUC (mmHg·h)	1972 ± 197	1879 ± 181	1947 ± 153	0.316	0.263
Pulse pressure, daily average (mmHg)	49 ± 7	47 ± 8	50 ± 8	0.236	0.231
Pulse pressure, 24-h AUC (mmHg·h)	1030 ± 154	987 ± 175	1056 ± 166	0.231	0.224
Heart rate, daily average (bpm)	78 ± 10	75 ± 10	73 ± 8	<0.001	0.303
Heart rate, 24-h AUC (bpm·h)	1634 ± 201	1580 ± 206	1528 ± 171	<0.001	0.316

Data shown as mean ± SD or means and 95% confidence intervals. P1 = *p*-value for end of study (week 36) against baseline (week 0). P2 = *p*-value for end of study (week 36) against end of fully-provided CRHP diet phase (week 6 or 12). CRHP: Carbohydrate-reduced high-protein; T2DM: Type 2 diabetes mellitus; BP: Blood pressure; AUC: Area under the curve.

## Data Availability

The authors commit to responsible sharing of data. This includes summary data and anonymized individual patient data as well as other material (e.g. protocols, clinical study reports). Requests from any qualified researchers who engage in rigorous, independent scientific research should be addressed to the corresponding author. Data will be provided following review and approval of a research proposal and statistical analysis plan and execution of a data sharing agreement.

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
