# Peer review of "Effects of a Self-Prepared Carbohydrate-Reduced High-Protein Diet on Cardiovascular Disease Risk Markers in Patients with Type 2 Diabetes"

_nutrients, 2021, doi:10.3390/nu13051694_

Round 1
Reviewer 1 Report
This paper describes the results from a 6-month open-label follow-up of a 12-week crossover trial that examined the effects of a fully prepared and provided carbohydrate-reduced high-protein (CRHP) diet on cardiovascular disease risk markers in subjects with type 2 diabetes. During the follow-up, all subjects self-prepared the CRHP diet with guidance from a dietitian. Beneficial effects on several cardiovascular risk factors that had been reported during the initial 12-week study were maintained during the 6-month follow-up and some markers showed additional improvements. The paper is very well written, and I have relatively few comments. However, one major point that I think deserves comment, and additional justification, is the choice to present the initial study results as the end of the fully provided CRHP condition, whether that was at the 6-week timepoint or the 12-week time-point and compare those results to the end of the open-label follow-up rather than to present the overall results at the end of the 12-week initial study and compare those to the end of the open-label follow-up. Additional suggestions are described below.
Specific Comments to the Authors:
- General comment: the paper includes many instances of 1st person writing (e.g., we and us). Suggest checking the journal guidance regarding whether 1st person is allowed, or if these should be rewritten in 3rd
- Consider adding a citation to the scientific statement on low-carbohydrate and very-low-carbohydrate diets for the management of body weight and cardiometabolic risk factors from the National Lipid Association Nutrition and Lifestyle Task Force (Kirkpatrick CF, et al. J Clin Lipidol. 2019;13(5):689-711.e.1.)
- Line 41: the global prevalence data described are for the change from 1980 to 2014. However, the reference cited [1] is a 2010 publication. Please check and correct this citation, and verify the reference numbering throughout.
- Lines 82 and 184: The 2 in m2 should be superscripted.
- Lines 92-93: suggest clarifying whether the dates shown of between April 2016 and November 2017 are the dates for the conduct of the initial 12-week study, or if this also includes the 6-month follow-up.
- Line 109: suggest changing “encouraged to reduce alcohol consumption” to “encouraged to limit alcohol consumption.”
- Line 118: was food intake recorded for all 7 days during weeks 19, 25, and 32? Or were these 3-day (or other) diet records?
- Line 149: suggest clarifying what “cheese 30+” means.
- Table 1: suggest adding percentages after the n’s shown for frequencies.
- Titles of Tables 2 and 3: the variables presented in Table 3 (e.g., blood pressure) would also be considered cardiovascular risk markers, thus suggest revising the titles of these tables to be more specific rather than labeling them generally as “CVD risk markers” and “cardiovascular function markers,” respectively. For example, Table 2 could be the effects on lipids, apolipoproteins, and inflammatory markers, and Table 3 could be the effects on blood pressure and heart rate.
- Footnotes of Tables 2 and 3: some of the abbreviations used in these tables do not appear in the footnotes, e.g., VLDL, NEFA, AUC, BP.
- Line 278: there appears to be an “x” missing between “2” and “6-week fully provided diet period.”
- Line 280: Suggest adding the mean baseline blood pressure values here.
Author Response
This paper describes the results from a 6-month open-label follow-up of a 12-week crossover trial that examined the effects of a fully prepared and provided carbohydrate-reduced high-protein (CRHP) diet on cardiovascular disease risk markers in subjects with type 2 diabetes. During the follow-up, all subjects self-prepared the CRHP diet with guidance from a dietitian. Beneficial effects on several cardiovascular risk factors that had been reported during the initial 12-week study were maintained during the 6-month follow-up and some markers showed additional improvements. The paper is very well written, and I have relatively few comments. However, one major point that I think deserves comment, and additional justification, is the choice to present the initial study results as the end of the fully provided CRHP condition, whether that was at the 6-week timepoint or the 12-week time-point and compare those results to the end of the open-label follow-up rather than to present the overall results at the end of the 12-week initial study and compare those to the end of the open-label follow-up. Additional suggestions are described below.
Reply: The initial part of the study was conducted in a cross-over design, where half of the subjects consumed the high-protein diet between 0-6 weeks, and the other half consumed the high-protein diet between 6-12 weeks. Hence, if the statistics and results were to presented in a pure chronological manner (e.g. 0, 12, 36), data at 12wk would be a mix of high-protein and control diets. Instead, we chose as midpoint the end of the fully-provided diet period, which was the 6wk time-point for half of the subjects, and the 12wk time-point for the other half. This way, changes during the 6 mo follow-up with the self-prepared high-portein diet represented changes from the end of the fully-provided high-portein diet. (otherwise they would represent changes from the end of the fully-provided control diet for half the subjects). We mention this rationale in the statistical analysis section (lines 167-173).
Specific Comments to the Authors:
- General comment: the paper includes many instances of 1st person writing (e.g., we and us). Suggest checking the journal guidance regarding whether 1st person is allowed, or if these should be rewritten in 3rd
Reply: We have checked the latest papers published in the Journal and verified that 1st person language is allowed (even in the abstract), e.g. https://doi.org/10.3390/nu13051409, https://doi.org/10.3390/nu13051408, https://doi.org/10.3390/nu13051413.
- Consider adding a citation to the scientific statement on low-carbohydrate and very-low-carbohydrate diets for the management of body weight and cardiometabolic risk factors from the National Lipid Association Nutrition and Lifestyle Task Force (Kirkpatrick CF, et al. J Clin Lipidol. 2019;13(5):689-711.e.1.)
Reply: We have added this in the revised manuscript (line 54; ref. 15).
- Line 41: the global prevalence data described are for the change from 1980 to 2014. However, the reference cited [1] is a 2010 publication. Please check and correct this citation, and verify the reference numbering throughout.
Reply: Thank you for spotting this. We have now inserted the correct reference (line 40; ref. 1).
- Lines 82 and 184: The 2 in m2 should be superscripted.
Reply: We have corrected this (lines 81 and 183).
- Lines 92-93: suggest clarifying whether the dates shown of between April 2016 and November 2017 are the dates for the conduct of the initial 12-week study, or if this also includes the 6-month follow-up.
Reply: Yes, these date are inclusive of the follow-up. We have clarified this in the revised manuscript (lines 91-92).
- Line 109: suggest changing “encouraged to reduce alcohol consumption” to “encouraged to limit alcohol consumption.”
Reply: Amended as suggested (line 108).
- Line 118: was food intake recorded for all 7 days during weeks 19, 25, and 32? Or were these 3-day (or other) diet records?
Reply: These were 3-day food records (2 weekdays and 1 weekend day). We have revised the manuscript to make this clear (line 116).
- Line 149: suggest clarifying what “cheese 30+” means
Reply: This notation is widely used for cheeses in Denmark and the Netherlands, and refers to the fat content of the cheese in dry weight. We have added this details in the revised manuscript (lines 147-148).
- Table 1: suggest adding percentages after the n’s shown for frequencies.
Reply: We added percentages for the frequencies shown in the revised Table 1.
- Titles of Tables 2 and 3: the variables presented in Table 3 (e.g., blood pressure) would also be considered cardiovascular risk markers, thus suggest revising the titles of these tables to be more specific rather than labeling them generally as “CVD risk markers” and “cardiovascular function markers,” respectively. For example, Table 2 could be the effects on lipids, apolipoproteins, and inflammatory markers, and Table 3 could be the effects on blood pressure and heart rate.
Reply: Revised the titles of Tables 2 and 3 as suggested (lines 216 and 228).
- Footnotes of Tables 2 and 3: some of the abbreviations used in these tables do not appear in the footnotes, e.g., VLDL, NEFA, AUC, BP.
Reply: Amended to include the missing abbreviation definitions (lines 220-221 and 232).
- Line 278: there appears to be an “x” missing between “2” and “6-week fully provided diet period.”
Reply: Indeed, we have now corrected this (line 277).
- Line 280: Suggest adding the mean baseline blood pressure values here.
Reply: We have added this information in the revised manuscript (SBP/DBP = 123/75) (lines 279-280).

Reviewer 2 Report
The study of Alzahrani et al describes very nicely the effect of a continuing carbohydrate-reduced high-protein diet in T2D patients with self prepared meals. These are very encouraging results showing that given the appropriate instructions, patients can retain the positive effect of a previously closely guided intervention.
However, given all the clinical data it would be excellent to add data concerning insulin, C-peptide and the calculated insulin clearance rate during the three time points (e.g. Rega-Kaun et al Obes. Surg. 2020; Bojsen-Moller KN et al J. Clin. Endo. Metab. 2013, Meier et al Am J Physiol Endocrinol Metab 2007).
Author Response
The study of Alzahrani et al describes very nicely the effect of a continuing carbohydrate-reduced high-protein diet in T2D patients with self prepared meals. These are very encouraging results showing that given the appropriate instructions, patients can retain the positive effect of a previously closely guided intervention.
However, given all the clinical data it would be excellent to add data concerning insulin, C-peptide and the calculated insulin clearance rate during the three time points (e.g. Rega-Kaun et al Obes. Surg. 2020; Bojsen-Moller KN et al J. Clin. Endo. Metab. 2013, Meier et al Am J Physiol Endocrinol Metab 2007).
Reply: We acknowledge this comment. Such data are indeed available, but in fact they were available several months earlier than the data presented in this manuscript. Biological samples were analysed for glucose, insulin and C-peptide before the second wave of COVID-related restrictions was put in place in Denmark (prior to December 2020). After restrictions were imposed in late December/early January, we had to cease all activities in the lab, so we could not analyse biological samples, retrieve subject raw data, etc., for the parameters presented in this manuscript (lipids, lipoproteins, inflammatory markers, blood pressure, etc.) until about a month ago, when restrictions were relaxed and we were allowed to resume activities in the lab. In the meantime, however, the glucose, insulin and C-peptide data were presented in another publication (which is currently in press).

Round 2
Reviewer 2 Report
It is very unfortunate that the authors did not provide access of their recently accepted manuscript during the submission process. I believe that the data is interesting but I do not agree with the ethical conduct of not declaring similar manuscripts during submission as I am not able to determine the novelty of the current work.
Author Response
We apologize for this misunderstanding. We were not aware we had to declare other manuscripts from the same project during the submission process. Clearly, we are aware of the strict policy about duplicate publications.
We have uploaded the other article we mentioned for your perusal. Obviously, the methods are similar since it is the same study, but the outcomes are totally different. In the previous paper we report on ectopic fat and glucose/insulin metabolic parameters, and none of the lipid metabolism and CVD parameters which are reported only in the manuscript submitted to Nutrients.
We have also provided the previous paper to the Academic Editor of the journal.
Thank you again for your review of our manuscript.
